# New Advances in Lateral Flow Immunoassay (LFI) Technology for Food Safety Detection

**DOI:** 10.3390/molecules27196596

**Published:** 2022-10-05

**Authors:** Guangxu Xing, Xuefeng Sun, Ning Li, Xuewu Li, Tiantian Wu, Fangyu Wang

**Affiliations:** 1Key Laboratory for Animal Immunology, Henan Academy of Agricultural Sciences, No. 116 HuaYuan Road, Zhengzhou 450002, China; 2College of Food Science and Technology, Henan Agricultural University, No. 95 WenHua Road, Zhengzhou 450002, China

**Keywords:** LFI, FRET, aptamer, SERS, QDs, electrochemical sensors, biosensors

## Abstract

With the continuous development of China’s economy and society, people and the government have higher and higher requirements for food safety. Testing for food dopants and toxins can prevent the occurrence of various adverse health phenomena in the world’s population. By deploying new and powerful sensors that enable rapid sensing processes, the food industry can help detect trace adulteration and toxic substances. At present, as a common food safety detection method, lateral flow immunochromatography (LFI) is widely used in food safety testing, environmental testing and clinical medical treatment because of its advantages of simplicity, speed, specificity and low cost, and plays a pivotal role in ensuring food safety. This paper mainly focuses on the application of lateral flow immunochromatography and new technologies combined with test strips in food safety detection, such as aptamers, surface-enhanced Raman spectroscopy, quantum dots, electrochemical test strip detection technology, biosensor test strip detection, etc. In addition, sensing principles such as fluorescence resonance energy transfer can also more effective. Different methods have different characteristics. The following is a review of the application of these technologies in food safety detection.

## 1. Introduction

Lateral flow immunoassay (LFI) is used in many applications, particularly in the field of food safety testing, including various pathogens, drug residues, food additives and other illegal additives, etc. [1,2,3]. The basic principle of LFI testing is the flow of a liquid test sample, with antibody-containing strips interacting with the analyte and discriminating the results by the accumulation of chromogenic substances. Although LFI is simple, inexpensive and portable, the results obtained are qualitative or at best semi-quantitative, which limits the wider use of LFI. In addition, the sensitivity of LFIs limits their use in trace detection and the accurate analysis of complex samples.

With the rapid and effective development of technology, in addition to addressing the shortcomings of LFI, researchers have combined techniques such as fluorescence resonance energy transfer (FRET), aptamers, surface enhanced Raman spectroscopy (SERS) and sensor with LFI, which has greatly contributed to the widespread development of LFI. This study reviews the current state of research over the past decade.

## 2. Fluorescence Resonance Energy Transfer (FRET)

FRET technology is based on energy transfer between two fluorophores without radiative excitation. Luminescent nanoparticles (NPs) can act as both donor and receptor emitters. The first fluorescent molecular material of the donor receives the energy and is excited, then converted to the second fluorescent molecular material of the acceptor. Thus, the molecular donor provides energy for the luminescent molecular material, and the receptor required for the luminescent molecular material to receive energy is located in the vicinity of 1~10 nm [4] as shown in Figure 1.

Tonsomboon [5] developed a fluorescent electrospinning fiber membrane that could bring mercury contamination to sub-ppb level within 60 s. The key to this user-friendly device lies in the development of a novel, organic NF06 mercury dye, which by exploiting the FRET mechanism, fluorescence enhancement-”turn-on” favorable results are presented, with a high Stokes shift (248 nm). The NF06 sensor was immobilized on cellulose acetate (CA)/polycaprolactone (PCL), and its surface chemistry was found to effectively enhance FRET operation. The results showed that the detection limit of the NF06 dye was 0.309 μg/L, which was lower than the mercury content allowed in water (maximum 2 μg/L) specified by the United States Environmental Protection Agency. Test strips without instruments were used to detect Hg^2+^ in drinking water and skin whitening serum, and the results indicated that the fluorescence of dipsticks was significantly enhanced when Hg^2+^ was present. These test strips are of low cost because they do not require expensive instruments and can therefore be used for general consumer safety assessments and various large-scale commercial screenings by regulatory agencies.

Bisulfate is essential in the body’s digestive function. Therefore, fast and simple detection of bisulfite is critical. Wen [6] reported a fluorescent probe (CPT) based on bisulfite (HSO^3−^) with a high Stokes shift (162 nm). HSO^3-^ can be quantitatively detected by this probe with a minimum detection limit of 45 nm, and it is highly specific compared to other commonly used anions and biothiols. The sensing mechanism is nucleophilic addition reaction and is corroborated by high resolution mass spectrometry. The probe paper is simple to make and easy to use.

Mycotoxins are toxic to animals, plants and humans, because mycotoxins survive thermal processing and hydrolysis processes, so detection of mycotoxins is critical. Oh [7] designed a FRET-based system for the detection of mycotoxins using antibody-antigen binding simultaneously, which can detect mycotoxins faster and more conveniently than other detection systems. In addition, the method can be attached to a nitrocellulose membrane, which allows the matrix effect in the sample to be more effectively counteracted, and can be used for the fluorescence measurement of Hematoxin A (OTA) in coffee samples. The system can complete the detection of OTA in 30 min and its detection limit (LOD) is 0.64 ng/mL. In the measurement results in the actually present samples, its LOD is 0.88 ng/mL, overcoming the matrix effect caused by the chromatographic properties of membrane capillary forces. Based on this, we can argue that the new system developed can be used as a powerful detection tool that can monitor food quality and environmental conditions, as well as sensitive diagnosis of harmful substances such as chemical reagents and microbial toxins.

## 3. Aptamer

Aptamers are oligonucleotide sequences with high specificity and a special affinity for target substances obtained by screening random oligonucleotide libraries by SELEX techniques. Compared with antibodies, aptamers have obvious non-immunogenicity, and they are low cost, high reproducibility, good stability and easily modify a variety of chemical groups. As shown in Figure 2.

Aflatoxin B (AFB1 and AFB2) frequently contaminates food, especially moldy cereal, posing a significant threat to people’s health; thus, it is necessary to study a sensitive, convenient and fast measurement method to prevent consumption of aflatoxin-contaminated food. Zhao [8] developed and designed a lateral flow adaptive sensor for detecting the aflatoxin B based on the application of the fluorescent dye Cy5 as an aptamer marker and competition between aflatoxin B and aptamer complementary DNA. This method is the first time that the aptamer complementary strand was used as the detection line (T line) for aflatoxin type-B. Furthermore, the authors also improved its affinity for aflatoxin B by using truncated aptamers and so the length of cDNA and aptamer probe was optimized to improve the sensitivity.

Therefore, binding organic solvent and buffers were also studied. The results indicated that a short aptamer (32 bases) combined with a probe complementing aptamer binding domain of the aptamer AFB1 was the best combination to improve AFB1 detection’s sensitivity and accuracy. Under the best experimental conditions, the linearity between bands of 0.2~20 ng/mL is good, and its detection limit is 0.16 ng/mL. The strip can be used for the detection of aflatoxin B in commercial almonds, peanuts and dried figs under certain conditions. The recoveries were 93.3–112.0%. The test strip has become a new type of detection tool, and will replace other detection tools in the future to detect aflatoxin B in moldy grains more quickly and sensitively, providing guarantees for food safety and helping people avoid eating food containing aflatoxin by mistake.

Researchers have designed a sandwich transverse flow strip detection method (LFSA). The content of rongalite in food was detected by AuNPs modified with gold nanoparticles [9]. More specifically, the biotin-labeled first-level A09 aptamer is fixed to the streptavidin envelope, and the second-level B09 aptamer binds to AuNPs as a capturing probe and a signaling probe, respectively. By recording the color changes in the LFSA control and test lines, the system can successfully directly detect aluminates in food samples at low concentrations of 1 μg/mL.

Zhang [10] proposed a new horizontal flow band fluorescence aptamer sensor based on a competitive format for one-step OTA determination in corn samples. In simple terms, the biotinylated cDNA is fixed to the surface of the nitrocellulose filter on the test line. In the absence of OTA, cy5-labeled aptamers bind to the complementary strand into a solid double-helix structure. However, under the conditions of an OTA, cy5-Aptamer/OTA complexes are produced, so less free aptamer is captured in the test area, resulting in a significant reduction in fluorescence signal on the test line. Its linear range is 1 to 1000 ng· ml^−1^, and the LOD was 0.40 ng mL^−1^, inhibitory concentration (IC_15_) was 3.46 ng/mL, and the recoveries were 96.4~104.67%. Therefore, in this work, the designed sensor can serve as a novel scheme to detect OTA, which can rapidly and sensitively detect OTA levels in particle samples.

Another study reports a preliminary study on the acquisition and function of a high-affinity DNA aptamers (DNA strands that specifically bind to the target DNA) coupled with a red-emitting quantum dot (Qdot 655) to improve the sensitivity of lateral flow (LF) bands for detecting food-borne pathogens. According to the literature, many scholars have researched and developed multiple DNA aptamers for the capture of Escherichia coli, Listeria monocytogenes and Salmonella enterica, and can also report their gene sequences. Importantly, research has showed that the strongest homologous bacteria could be identified using the colloidal gold screening system detection. The results showed that several promising sandwich combinations were found in three bacterial LF band systems. By further studying the best LF system, (LOD) of about 3000 *E. coli* (8739) and about 6000 *E. coli* (O157: H7) was obtained. Their LODs were decreased to about 300–600 bacterial cells separately in each test, due to changing to the Qdot 655 APTAMER-LF system. There are new changes to these tests, for instance using a high level of detergent to avoid QDs gathering; and enhanced membrane migration, the determination of optimal analysis of membrane type ultraviolet immobilized capture adapter body, discussing the new double biotin/tag at the end of the digoxin nucleic acid body chain, mildew avidin colloidal gold or Qdot 655 coupling line with anti-digoxin antibody. In conclusion, this work offers a proof of theory for rapid detection of foodborne pathogens with highly sensitive aptifier Qdot LF bands [11].

## 4. Surface Enhanced Raman Spectroscopy (SERS)

SERS is a new technology based on molecular detection and characterization. It depends on the enhanced Raman scattering of molecules near the hot spots of a SERS active surface. Spectroscopy or Raman scattering has become a non-destructive testing technology. Its excellent characteristics are sensitivity, rapidity, accuracy, fingerprint information and so on. Raman scattering is an inelastic scattering, which is a kind of scattering between media and photons. The spectrum can reflect the structural characteristics of the investigated samples. For heavy metals such as gold and silver, the enhancement effect of rough surface is called the SERS effect. Wang et al. [12], based on the working principle, strategy and point-of-care testing (POCT) application of the SERS test strip, designed horizontal and vertical flow test strips. SERS-based POCT and field analysis test strips need to be equipped with a portable high-power Raman spectrometer equipped with a high-power narrow-spectrum Raman bandwidth laser, so as to transform laboratory analysis into practical analysis applications, as shown in Figure 3.

β-Conglycinin is an important nutrient substance in soybeans, and eating too much may cause food allergies. Xi et al. [13] studied a sandwich lateral flow immunochromatography detection strip, which was mainly established by high-affinity mouse monoclonal antibody (3D11 mAb), to quickly and accurately detect soybean allergen β-Conglycinin. 3D11 mAb binds to a rabbit polyclonal antibody to form a band. The LOD of this test strip was 1 μg/mL. In addition, the quantitative detection of β-glycine was achieved by using the chemical molecules p-aminothiophene and colloidal gold as Raman-enhanced signals. This study determined that β-glycine concentrations work between 160 ng/mL and 100 μg/mL.

Based on the cheap, simple, fast and portable LFI strip and SERS technology, an ultra-sensitive detection method for antibiotics in milk was established. Immunoprobes are prepared by colloidal gold (AuNPs) with anti-neomycin (new monoclonal antibody) and coating competition between the Raman antiprobe molecule 4-aminothiol (patient) and free NEO and antigen (NEO-OVA), resulting in changes in the number of immunoprobes immobilized on the paper-based material. To quantify NEO, the Raman intensity of PATP on the lateral flow test line needs to be determined. The LOD and semi-inhibitory concentration values were 0.216 pg/mL and 0.04 ng/mL, respectively. This method does not cross react (CR) with other compounds and has high specificity. Recoveries of milk added to new samples ranged from 89.7% to 105.6%, and a relative standard deviation (RSD) of 2.4–5.3% (n = 3). The results showed that the proposed method has high sensitivity, specificity and stability, and can detect various antibiotic residues added in milk samples [14].

Kim et al. [15] developed a metal-organic framework (MOF) coating-based SERS paper-based platform for detecting cadaverine and putrescine. Gold @ Zeolite-Imidazole Skeleton-8 (ZIF-8) SERS paper-based preparation was accomplished by coating a ZIF-8 layer on gold nanoparticle-impregnated paper, and paper-based devices were prepared by the dry plasma reduction method. The study characterized the material by scanning electron microscopy. The ZIF-8 coating concentrates gas molecules and enhances the SERS signal. Fluorescence, SERS and simulation results show that this method improves the detection capability of the Au@ZIF-8 platform. To be able to efficiently and selectively detect putrescine and cadaverine, the Au@ZIF-8 SERS paper-based device was functionalized with 4-mercaptobenzaldehyde (4-MBA). The 4-MBA molecule can replace the Raman reporter gene and is also a specific receptor for volatile amine molecules. The paper-based system can quantitatively detect putrescine and cadaverine, their detection limits are 76.99 μg/L and 115.88 μg/L. In addition, the detection of volatile amine molecules released from spoiled pork, beef and chicken samples was proved. The SERS paper-based platform for predicting MOF coating is not only suitable for food quality testing, but also for other aspects such as environmental monitoring and clinical diagnosis.

Huanhuan et al. [16] designed a SERS-based rapid detection of chloramphenicol in food samples that deposited floral AgNP in the hydrophobic region of cellulose paper to prepare flexible SERS sensors for obtaining SERS signals from chloramphenicol (CAP). A study designed SERS-based AgNPs test strips that reacted with ingredients in white or red wine [17]. Flavor substances and color are rapidly accompanied and captured by AgNPs to generate specific SERS signals for sensitive and accurate detection of different compositions in wine. Verma [18] used a simple in situ one-step silver mirror reaction (SMR) to grow silver nanostructures on SERS substrate filter paper to prepare a paper-based SERS substrate to detect chemical pollutants in milk and water: urea and melamine. Label-mediated detection involves applying antibodies or aptamers specifically binding to the microorganism under testing, which can improve the accuracy and sensitivity of the assay. Xu et al. reported a Salmonella Typhimurium detection method using DNA-assembled gold nanodimer for typhimurium aptasensor, which has been successfully tested in labeled milk with an LOD of 35 cfu/mL [19].

With the advancement of modern science and strict requirements for food safety, the detection limits of various substances are required to become lower. Therefore, the detection requirements of pesticide residues are also getting higher and higher. Research in recent years has shown that the detection of trace pesticide residues must be achieved through sensitive detection technology. Xu et al. have applied SERS technology to pesticide detection [20]. Sheng et al. [21] detected imidacloprid and offlufen based on SERS. The antibody is combined with SERS nanolabels, and a surface-enhanced SERS-based translateral flow analysis (SERS-LFA) is prepared simultaneously using competitive immunobinding. Nanoribbon catalytic strips were developed for pesticide SERS detection in sewage, equipped with a thin layer of plasma gold (Au @ BiOI) for TCP detection. SERS detects an impressive enhancer factor of TCP 10^7^ with a LOD of 10^−10^ [22].

Sensitive quantitative immunoassay of hydrophobic SERS supercritical fluids with improved lateral flow rods. Hydrophilic silver nanoparticles were sputtered to the specific surface area of hydrophobic polydimethylsiloxane by magnetron sputtering, and hydrophobic polymer bands with Raman internal standards were prepared. The analyte of interest can be enriched in test lines and control lines formed in the hydrophilic Ag region, and the sensitive quantitative monitoring of ultrafast-band microferrin (FER) based on fluorine immunoassays significantly improves the adaptability of SERS [23]. An aptamer-functionalized SERS active silver-branch crystal matrix for the detection of food allergens on food contact surfaces was developed by Boushell et al. This work improved the method for detecting model food allergen lysozyme in samples gathered from food contact surfaces at a minimum concentration of 5 μg/L [23].

Liu et al. [24] used Fe_3_O_4_@AuNPs not only as a tool for isolation and purification, but also as SERS tags for quantitative analysis. With this strategy, dipsticks can be quantitatively analyzed for targets in untreated blood samples. C reactive protein (CRP) and serum amyloid A (SAA) had detection limits of 0.01 ng/mL and 0.1 ng/mL, respectively. By using the same antibodies, the LOD was 100 times and 1000 times more sensitive than the standard colloidal gold dipsticks. Pesticide residues in juices and water can be detected using SERS-active substrates prepared from textile fibers with AgNPs and liquid crystal polymers (LCPs). The filter membrane-based SERS substrate preconcentrates the analyte in situ by continuous filtering and thus improves the quality of the SERS signal of interest [19]. Li et al. [25] developed a SERS-integrated LFS platform that can quickly and simultaneously screen for many varieties of genetically modified organism (GMO) components (promoters, codons and terminators) in legumes. If these multiple analytes coexist or vary with the same concentration trend, these multiple good manufacturing practice (GMP) components can be accurately and quickly screened (15 min) by reading the signal on the same test line.

At present, SERS technology is widely used in the field of food safety testing, but there are still some disadvantages. In order to make SERS technology a basic tool for food quality safety and environmental monitoring, future research should combine SERS with other technologies to develop new SERS substrates, improve experimental protocols and reduce costs.

## 5. Quantum Dots (QDs)

“Quantum dots (QDs)”, conjugated with high-fluorescence probes, are essential for detection and long-term fluorescence imaging of some kinds of cellular processes [26,27]. QD is one of the most promising immunochromatography markers. The quantum dots used have different diameters (1–100 nm); it is possible to generate luminescence covering the entire visible region through a single near ultraviolet excitation source [28,29]. It is worth noting that each quantum dot is characterized by a narrow emission spectrum. Hence, quantum dots can be used as novel molecular imaging probes, and are also considered a remarkable reagent against viral infection. In addition, collaboration on potentially biocompatible vectors could help with interdisciplinary research and enable clinical approaches to combat viruses, so the function of QDs can be as carriers/labeling drugs or drug carriers. As shown in Figure 4. Multicolor QDs have been used in a variety of fields, including biological imaging [30], fluorescence-based ELISA [31,32], flow cytometry [33], etc.

Studies have shown that an immunochromatographic test was developed to simultaneously detect several species in complex sample substrates. This work was designed in the format of “traffic lights” and three different colored lines are on the test strip, therefore, a simple and easy-to-use tool has been developed that can identify and analyze substances according to the different colors of existing lines (qualitative analysis) and determination of the amount of analyte according to the fluorescence intensity of the test line under UV light (quantitative analysis). To develop multicolor immunochromatography tests, the antibody can selectively bind to these three different types of antibiotics, such as ofloxacin, chloramphenicol and streptomycin. The quantum dots of each antibody have an emission maximum wavelength (525, 585 or 625 nm). The detection limits for these three antibiotics were 0.3, 0.12, and 0.2 ng/mL. The data obtained from the study revealed values 80–200 times lower for the ELISA used (same antibody). Analysis using this system requires no additional sample pretreatment, is simple to operate, and can detect these antibiotics in milk samples in a short time, by just dropping the milk into the area to be tested. The method assay also showed high analyte detection (92–101%) and accuracy (quantitative error <8% of the mean) to detect added milk samples [34]. Also developed was a lateral flow immunoassay technique for the competitive, fluorescent detection of triclopyridine (a metabolite of organophosphorus insecticide poisoning tick phosphorus) by QDs in rat plasma [35], which can also detect Mycotoxins-Ochratoxin a in red wine and antibiotics (chloramphenicol) in milk [36,37].

Another study describes the first quantum dot-based test paper nucleic acid biosensor. Briefly, the reported lateral flow of streptavidin-functionalized CdSe-Zns core-shell QDs was used to detect. Quantum dot-based nucleic acid biosensors can visually detect PCR amplification products and polymerase chain reaction (PCR) amplification products, and visual genotyping of deoxyribonucleotide single nucleotide polymorphisms (SNPs) in the human genome. No purification steps are required before a DNA sample is applied to a type I or II sensor strip. Double-stranded DNA down to 1.5 fmol can be clearly detected with the naked eye, with dynamic range extending to 200 fmol. Its coefficient of variation was estimated to be 4.3–8.2 [38].

The heavy use of ciprofloxacin (CIP) has caused a number of health threats [39,40]. Hence, many nations and organizations have determined the maximum residue limits (MRLs) of CIP; as stipulated by the European Commission and the Ministry of Agriculture and Rural Affairs of the People’s Republic of China, the maximum residue limit of CIP in fish shall not be more than 100 μg/kg. In order to meet the needs of rapid detection on agricultural planting, production and harvesting sites, such as good stability, high accuracy, low cost, wide range of use, etc., the quantum dot microsphere (QDM)-based immunochromatographic quantitative CIP test strip and the complete CIP detecting solutions including an intelligent test strip reader was designed [41]. When the sample to be tested contains CIP, since the QDM-monoclonal antibody (mAb) probe can bind to CIP, it cannot be captured by the CIP-bovine serum albumin (BSA) conjugate dispersed on the T line, thereby decreasing fluorescence strength. The linear detection range of the test strip obtained in this study was 0.1–100 ng/mL, and the LOD was 0.05 ng/mL, which has the characteristics of high sensitivity, good stability and high accuracy. Furthermore, in order to automatically quantitatively detect CIP, this study used 3D printing to manufacture a smartphone test strip reader with dimensions of 85 mm × 48 mm × 44 mm. The entire process of CIP testing can be completed in less than 15 min, but it only costs about 1 yuan (10 cents).

The fluorescence immunochromatographic banding test (ICST) based on QDs has been exploited and used to detect fumonisins in maize samples [42]. Based on the extraction process, the detection limit of fumonisin was 2.8 µg/L, and the linear range of analysis was 3–350 µg/L, which is equivalent to 30–3500 µg/L of corn meal samples. Its sample preparation and analysis time is a total of 22 min. The recoveries of enhanced and naturally contaminated corn meal samples ranged from 91.4%–105.4%, and the coefficient of variation was less than 5%. To evaluate the potential improvements that QD could bring to ICST technology, the researchers directly compared the proposed QD-ICST with gold nanoparticles and chemiluminescence ICST for fumonisin detection using the same immunoreagents.

## 6. Electrochemical Sensor Strips

While significant advances have been made in sensing technology, the design, installation, and application of self-test equipment remains limited to users and specialized laboratories. Decentralized analytical equipment is very helpful for people’s production and life, so that people can easily analyze, evaluate and predict various environmental and food samples, which can greatly improve people’s living conditions, especially for people living in remote or resource-poor areas [43]. In recent years, there has been a growing focus on paper-based analysis tools. As we all know, paper has many excellent properties, including portability, environmental friendliness and low cost, coupled with the ability to print electrochemical sensors, so that there are more opportunities for sustainable equipment that can drive (bio) sensors beyond the most advanced level. As shown in Figure 5. These devices are also suitable for non-specialists because they are easy to carry and easy to operate and easy to learn [44].

Ding [45] outlined the role of paper-based potentiometric and voltammetry in heavy metal detection. Paper-based 3D sensors are made by bonding and folding different layers of a piece of paper, and the objects of its use depend on the type of sensor, for example it can be used for chromatography, electrochemistry and colorimetric processes [46]. Nguyen [47] introduces the latest developments in high-sensitivity electrochemical paper-based lateral-flow assays (LFAs), which has many unique advantages because of the versatility and simple optical setup of electrochemical LFA.

Zhang [48] used benzoquinone (BQ)-mediated E. coli respiration to develop a new microfluidic paper-based analysis device (μPAD) for detecting the biotoxicity of contaminants. The detection of K_3_ [Fe(CN)_6_] by cyclic voltammetry confirmed the good electrochemical properties of the paper-based carbon three electrodes. Liu [49] developed paper electrochemical devices (PEDs), which is low cost, portable, and easy to use. It can detect a wide range of small molecule analytes in complex sample matrices on the spot. Pungjunun [50] uses an origami design to create a disposable gas-sensitive paper-based device (gPAD) that integrates a gas adsorbent and an electrochemical detection area into one device, gPAD, for detecting the NOx gases uses screen-printed graphene electrodes to modify copper nanoparticles (CuNP/SPGE), which increases the sensitivity of the device and improves selectivity. Li [51] has developed a paper-based biochip that triggers H_2_O_2_ cleavage fluid switch-mediated multiplication and quantification through visual screening and a dual-response output of radiometric electrochemistry for field sensitive detection of targets. Another study describes an electrochemical paper-based analytical device (ePAD) drawn with graphite pencils to determine ascorbic acid (AA) in commercial tablets. Under the best conditions, the new device developed for the quantitative determination of glucose has a linear range of 0.1 mM to 25 mM and a detection limit of 25 μM [52].

A paper-based 3D sensor is bonded and folded from different layers of paper. Because it combines multiple preparation means, integrates different sensor components, and connects two detection methods, it is more user-friendly in small collections. The object of use of a paper sensor depends on the type of sensor, e.g., it can be used for chromatography, electrochemistry and colorimetric processes. In addition, in recent years, these sensors have been studied extensively and applied in many aspects, such as food safety detection, disease diacrisis and environmental monitoring. Another study describes a paper-based electrochemical sensor based on a thiol-terminated poly(2-methacryloyloxyethylphosphorylcholine) (PMPC-SH) that self-assembles on a screen printed electrode (SPE) modified with gold nanoparticles [53]. Another study uses lithography and screen-printing technology to build a new 3D paper-based microfluidic SPE. This paper-based, three-dimensional microfluidic electrochemical biosensor can quantitatively detect glucose [54]. Panraksa [55] has developed a convenient paper-based sensor that can quickly determine acetylcholinesterase (AChE) with high selectivity and sensitivity, in which screen-printed graphene electrodes are used as working electrodes to increase the sensitivity of the sensor.

## 7. Biosensor Strips

Biosensors are small devices that convert the recognition of biomolecules (DNA/RNA, antibodies, proteins, whole cells, etc.) into signals (electrochemical, optical, piezoelectric, nanomachinical, mass sensitive, etc.) and are considered powerful tools for detecting target molecules or products in environmental and food samples. Biosensors are divided into many types, among which fluorescence and colorimetric sensors are relatively simple and intuitive to observe the number of samples, and can determine the sequence of target DNA through fluorescence. Biosensors play an important role in food detection, with many advantages such as portability, specificity, low cost and short time, which can simplify and automate detection methods [56]. As shown in Figure 6.

A phage-based dipstick biosensor has now been developed and tested to detect a variety of foodborne pathogens in food samples [57]. A four-target nitrocellulose-based lateral flow immunoassay biosensor has also been developed in a form of dry reagent strip to interpret double-labeled double-stranded amplicons in thermally stable triple-ring-mediated isothermal amplification analyses [58].

RAgamOBP1 (Odor Binding Protein OBP of Anopheles anopheles gambiana) is appropriate for “dipstick” biosensors with portability, low cost, etc., improving lateral flow technology because insect OBP is robust, easy to obtain by recombinant expression, and resistant to detector “contamination” [59]. Melnik [60] designed a lateral-flow immunoassay (dipstick) that uses plant-derived antibodies to sensitively detect [Arg4] microcystins at concentrations of 100–300 ng/L in freshwater samples taken in different regions.

Immunological approaches are among the tools that have the opportunity to develop easy-to-manipulate biosensors. Vitamin B12 in food is one such analyte that needs to be urgently tested. This study describes a dipstick-based immunochemiluminescent (immuno-CL) biosensor for detecting vitamin B(12) in energy-replenishing beverages [61]. This is a form of direct competition that involves fixing vitamin B(12) antibodies to nitrocellulose membranes (NC) and then treating them with vitamin B(12) and vitamin B(12) alkaline phosphatase complexes to promote competitive binding. After system optimization, its detection limit is 1 ng mL^−1^. This study performed a sensitive, accurate and rapid screening of vitamin B (12) in a beverage sample capable of replenishing energy. Immune CL-based dipstick technology is appropriate for detecting a variety of targets in food and environmental samples.

The surface plasmon resonance (SPR) biosensor is an optical biosensor that is widely used in food inspection. It is an optical system capable of exciting and interrogating surface plasmons, as well as having biomolecular identification elements for detecting and capturing the analyte of interest present in the sample. The optical signals discover the binding analyte on the identification element, and it can change the surface plasmons propagation constant and lead to changes in the surface refractive index. SPR helps to detect many ingredients in foods such as antibiotics, chemicals and microbial toxins without labels and to ensure safety [62]. Unlike other biosensors, SPR-based biosensors, which are combined with the receptor assay, pave an innovative path for detecting antibiotics in food [63]. Peng et al. [64] developed an SPR-based immunosensor with a monoclonal antibody, 5B10, for detecting the presence of MQCA (3-methyl-quinoxaline-2-carboxylic acid), the signature residue of Olaquindox, in edible pig tissues. This method of SPR proved to be more efficient than the indirect competitive ELISA technique in terms of its label-free nature, LOD, sensitivity, accuracy and precision.

Many researchers use SPR to test for adulteration in milk, baby food, pet food, pure honey, tequila and alcoholic beverages and meat sausages. Vikas et al. [65] studied the effect of adding a Ag film fiber-optic SPR probe to increase the sensitivity of glucose and fructose in adulterated honey samples to 24% and 37%, respectively. Houhoula et al. [66] investigated the adulteration of the horse cytochrome B gene in meatball samples by observing fluctuations in the wavelength of the acid-adding solution. The addition of acid limits the agglomeration of gold nanoparticles in the sample, resulting in a pink color. The unadulterated sample is purple and the maximum absorption wavelength is 524 nm. The peak wavelength of the adulterated sample solution is gt. 570 nm. Li et al. [67] developed a molecular imprinting-based probe. Melamine was characterized by Fourier-transform infrared spectroscopy (FTIR) and atomic force microscopy (AFM). SPR-enhanced DNA biosensors and sandwich DNA detection technology were used to detect sausage meat adulterated with donkey meat. This study proved to be highly selective and specific for complementary genes, as well as for mismatches between cooked sausages and sausages adulterated with donkey meat [68]. In another study, it was observed that, with the adulteration of sugars such as glucose and fructose in pure honey samples, the incidence angle of this SPR sensor decreased with the increase of the adulteration percentage of the transition SPR curve at its peak [65].

## 8. Conclusions and Future Scope

Detecting adulteration and toxic substances in samples requires powerful sensors with ultra-high reproducibility. These sensors ought to be dependable and mass-manufactured to manage more and more samples for mass screening. The importance of LFI in the detection of food contaminants and adulteration was introduced in this paper (Table 1). The advantages and disadvantages of FRET, Aptamer, SERS, QDs, electrochemical test strip detection technology, the biosensor dipstick test, immunochromatographic test strips and other methods were analyzed. Each approach using electrical, optical and electrochemical techniques was elaborated and the concrete challenges were discussed. Food testing technology, combined with test paper into portable equipment, is an important assignment to implement these sensing tactics as a subversive technology in a real-world environment, linking research in the lab and the industry. Sensors should be rugged and should bear interference from a variety of environmental factors to adapt to the equipment available in the field. Further advances in smartphone-assisted sensing, real-time data analysis and device miniaturization will accelerate the adoption of these sensing strategies. Given all that has been discussed in this article, many exciting ways in which LFI will grow mature into a practical device for detecting food adulteration and toxic substances in the immediate future can be imagined.

## Figures and Tables

**Figure 1 molecules-27-06596-f001:**
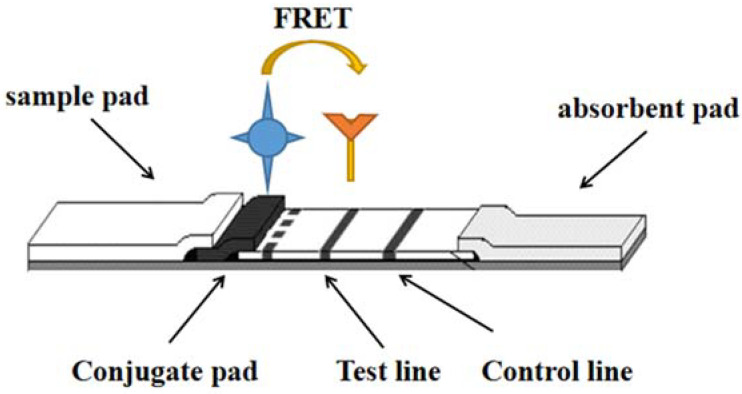
One of the detection principles of LFI–FRET.

**Figure 2 molecules-27-06596-f002:**
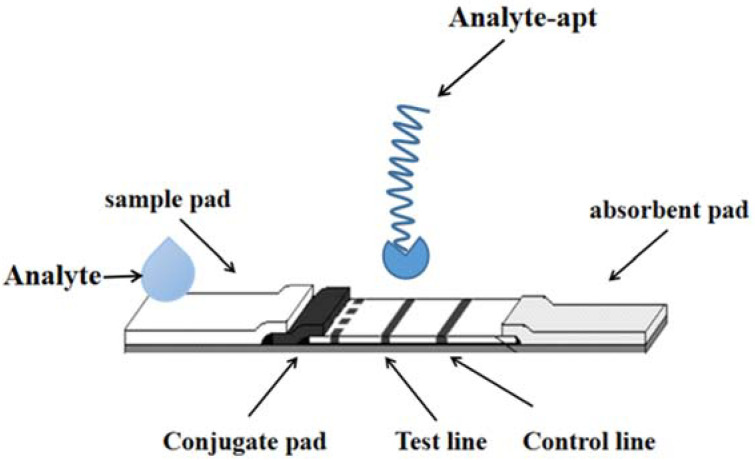
The detection technology of LFI–Aptamer.

**Figure 3 molecules-27-06596-f003:**
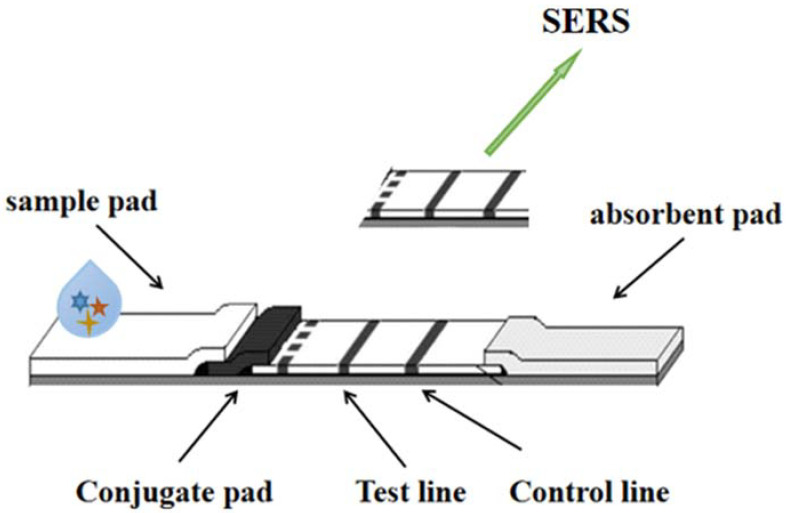
The detection technology of LFI–SERS.

**Figure 4 molecules-27-06596-f004:**
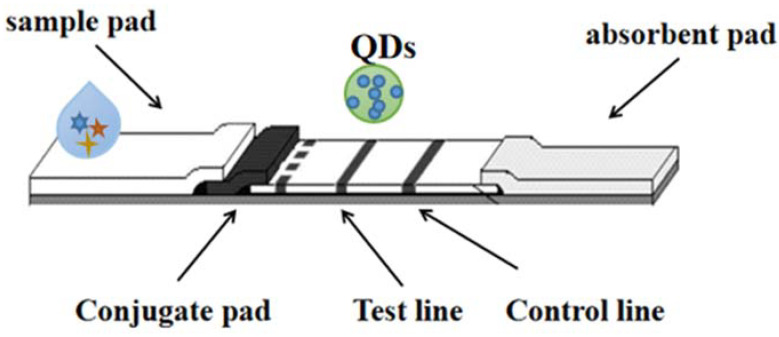
The detection technology of LFI–QDs.

**Figure 5 molecules-27-06596-f005:**
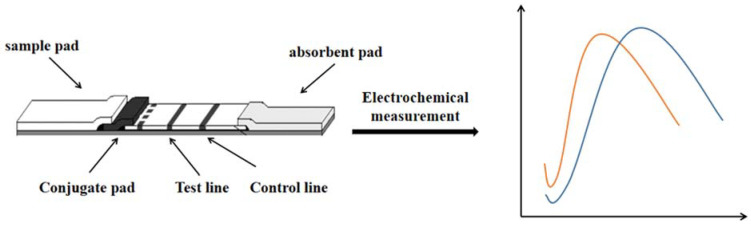
The detection technology of LFI–Electrochemical sensor.

**Figure 6 molecules-27-06596-f006:**
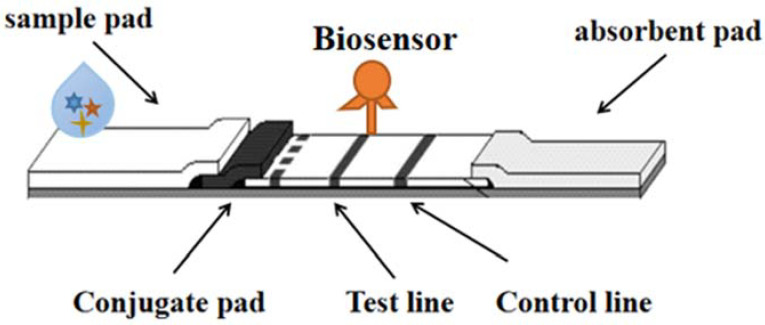
The detection technology of LFI–Biosensor.

**Table 1 molecules-27-06596-t001:** Literature summary on LFI.

Type of Detection	Analyte	LOD	Analyzed Samples	Reference
FRET	Hg^2+^	0.309 μg/L	water, skin whitening serum	[5]
FRET	HSO_3_^−^	45 nM	cell	[6]
FRET	Hematoxin A	0.64 ng/mL	coffee	[7]
Aptamer	Rongalite	1 μg/mL	food	[8]
Aptamer	aflatoxin B	0.16ng/mL	almonds, peanuts, dried figs	[9]
Aptamer	Hematoxin A	3.46 ng/mL	corn	[10]
SERS	β-Conglycinin	1 μg/mL	Skimmed milk	[13]
SERS	antibiotics	0.216 pg/mL	milk	[14]
SERS	Cadaverine, putrescine	76.99 and 115.88 μg/mL	spoiled pork, beef, chicken	[15]
SERS	S.typhimurium	35 cfu/mL	milk	[19]
SERS	C reactive protein erum amyloid A	0.01 and 0.1 ng/mL		[23]
QDs	antibiotics	0.3, 0.12, 0.2 ng/mL	milk	[34]
QDs	ochratoxin A	1.9 ng/mL	Red win	[35]
QDs	ciprofloxacin	0.05 ng/mL	fish	[40]
QDs	fumonisins	2.8 µg/L	corn	[41]
Electrochemical sensor	*E. coli*	13.5 μg/mL	cucumber	[47]
Electrochemical sensor	C-reactive protein	1.6 ng/mL	human serum	[52]
Electrochemical sensor	glucose	25 μM	human sweat and blood shows	[53]
Electrochemical sensor	acetylcholinesterase	0.1 U/mL	blood	[54]
Biosensor	foodborne pathogens	10–50 CFU/mL	spinach, ground beef and chicken homogenates	[57]
Biosensor	odorant-binding protein	100 mg/L	water	[59]
Biosensor	[Arg4]-microcystins	12.5 ng/L	water	[60]
Biosensor	Vitamin B_12_	1 ng/mL	energy drinks	[61]
Biosensor	3-methyl-quinoxaline-2-carboxylic acid	1.4 µg/kg and 2.7 µg/kg	swine muscle and liver	[64]
Biosensor	glucose and fructose	5.67 × 10^−4^ and 2.9 × 10^−3^ RIU	honey	[65]
Biosensor		12.3 fg/μL	horse meat	[66]
Biosensor	Melamine	2.5 mg/L		[67]

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
