# Peer review of "New Advances in Lateral Flow Immunoassay (LFI) Technology for Food Safety Detection"

_molecules, 2022, doi:10.3390/molecules27196596_

Round 1
Reviewer 1 Report (New Reviewer)
This manuscript is well written and well organized. In my opinion, it can be accepted after the following revisions to strengthen this manuscript:
1. The image sent has illegible letters, the authors must modify the font size and the images.
2. The authors must add in each section a figure with the images of the works explained within each section.
3. The units must be unified in the international system (ppb or ng/mL or µg/L, as appropriate).
4. The authors should add a table with different published works of each type of detection, in addition to those commented on in the main text. The type of detection, analyte, linear range, detection limit and analyzed samples must appear, in addition to the reference.
Author Response
Dear Friend:
Thank you very much for your careful review of my manuscript. I have revised all the issues according to your comments. A sentence-by-sentence response to the revisions is attached.
Thank you very much.

Reviewer 2 Report (New Reviewer)
Immunochromatography is a very useful technique for detecting small and large molecules in many fields, such as medical diagnosis, food safety detection, environmental monitoring and so on. Therefore, the latest research progress of ICA technology can be reviewed in time, which is a very meaningful reference for scientists. However, there are still some deficiencies in the writing of the manuscript which need to be carefully improved. The two main problems are as follows:
1. The manuscript lacks the necessary representative graphic abstracts or technical schematic diagram for the selected techniques, which makes it slightly less readable. To help readers better understand the content, Pls add them.
2. The logical structure of the manuscript content can be further improved. For example, are technologies classified according to principles, probes, or others? Are there type specific subtypes under each category? Please make improvements as appropriate.
Author Response
Dear Friend:
Thank you very much for your careful review of my manuscript. I have revised all the issues according to your comments. A sentence-by-sentence response to the revisions is attached.
Thank you very much.

This manuscript is a resubmission of an earlier submission. The following is a list of the peer review reports and author responses from that submission.
Round 1
Reviewer 1 Report
This article reviews the latest development of lateral flow immunoassay in food safety testing. However, after carefully reading the paper, I did not find the corresponding summary and prospects for future development. In addition, the logical framework of this review is not clear, and the literature classification is confused. Besides, I suggest that the author add corresponding tables and figures to summarize the latest progress in this field, thus helping the readers better understand the latest development of this field.
Reviewer 2 Report
This manuscript reviews a number of research articles that use lateral flow immunochromatography (LFI) method for food safety testing, environmental testing, and clinical medical tests. New technologies are discussed that combined with the LFI method, increase capabilities; such new technologies are like the use of fluorescence resonance energy transfer, aptamer, surface enhanced Raman spectroscopy, quantum dots, electrochemical test strip detection technology, and biosensor dipstick tests. Several studies have been reported in this manuscript in each category, and in each case some characteristics are reported.
The topic is interesting and sensing in the food industry have very many interests for the researchers, and at the same times in is extremely vital in the industrial applications.
As for the present manuscript, I consider it as a nice collection of summaries; it reports many interesting articles, that is required for a review paper. However, this manuscript lacks significantly the element of “Novelty”; another required aspect to be considered for publication in the journal of biosensors. The “Conclusion” is also very much weak, and requires improvements.
Given that, I suggest the authors to consider revising the manuscript significantly, before it is ready to be published as a review paper. The following is a list on the points I suggest to consider and improve:
As I mentioned, the manuscript for the most part is a collection of summaries; since they are summarized one by one and without strong synthetic comparison, it’s not straight-forward to realize the advantages / disadvantages of one method over the others, working conditions or limits. First of all, some diagrams to better illustrate each method could be very much appreciated. Moreover, the authors might consider to add a table with all the characteristics of each method (including limit and range of detection, etc). Doing this way, many parts in the text could be shortened and the information can be moved to such a table.
Since there are recently lots of attention on the use of 2D nanomaterials (for instance, for their very high surface to volume ratio), the authors could present the literature and their view on the use of this class of materials on the sensing devices. This is essential for an up-to-date review. Examples of such innovative materials and their application in sensing and bio-related fields are:
RSC Adv., 7, 50166-50175 (2017)
Sensors & Actuators: B. Chemical 320, 128440 (2020)
Anal. Chem., 94, 24, 8693–8703 (2022)
Another improvement could be adding a discussion for a possible use of the photonic immobilization technique, also used in: Applied Physics A volume 117, pages 185–190 (2014) to improve sensitivity of the sensor; this in fact could very likely increase performance and sensitivity of the LFI sensors.
Other points:
- The last sentence of the first paragraph; “In addition, the sensitivity of LFIs limits their use in trace detection and the accurate analysis of complex samples.” need more explanation and references.
- page 2, paragraph 3: What “CPT” stands for? it has to be mentioned.
- page 3, paragraph 3: What do you mean by “AuNPs modified with gold nanoparticles” ???
- page 3, paragraph 5: we read “Their LODs were decreased to about 300-600 bacterial cells”. Rather than the term “decreased”, it is advisable to use the term “improved”, for instance.
- page 4, paragraph 3: What is “NEO”?
- page 5, paragraph 5: What is “LFS”?
- page 9, paragraph 2: we read “ … the effect of adding a layer of go to a silver-plated fiber-optic SPR probe to … ”. What is “a layer of go” ??
- page 9, paragraph 3: we read “ … methods were analyzed … ”. The analysis is very poor and has to be certainly improved.